# Regulatory Crosstalk between Physiological Low O_2_ Concentration and Notch Pathway in Early Erythropoiesis

**DOI:** 10.3390/biom12040540

**Published:** 2022-04-02

**Authors:** Véronique Labat, Eva Nguyen van Thanh dit Bayard, Alice Refeyton, Mathilde Huart, Maryse Avalon, Christelle Debeissat, Laura Rodriguez, Philippe Brunet de la Grange, Zoran Ivanovic, Marija Vlaski-Lafarge

**Affiliations:** 1R&D Department, Etablissement Français du Sang Nouvelle-Aquitaine, 33075 Bordeaux, France; veronique.labat@efs.sante.fr (V.L.); nguyeneva@hotmail.fr (E.N.v.T.d.B.); alice.refeyton.ext@efs.sante.fr (A.R.); mathilde.huart@u-bordeaux.fr (M.H.); maryse.avalon@efs.sante.fr (M.A.); christelle.debeissat@u-bordeaux.fr (C.D.); laura.rodriguez@efs.sante.fr (L.R.); philippe.brunet-de-la-grange@efs.sante.fr (P.B.d.l.G.); zoran.ivanovic@efs.sante.fr (Z.I.); 2Inserm Bordeaux UMR 1035, 33000 Bordeaux, France; 3Université de Bordeaux, 33076 Bordeaux, France

**Keywords:** erythropoiesis, CD34^+^ cells, HIF, Notch, progenitors, BFU-E, CFU-E, low O_2_ concentration, differentiation

## Abstract

Physiological low oxygen (O_2_) concentration (<5%) favors erythroid development ex vivo. It is known that low O_2_ concentration, via the stabilization of hypoxia-induced transcription factors (HIFs), intervenes with Notch signaling in the control of cell fate. In addition, Notch activation is implicated in the regulation of erythroid differentiation. We test here if the favorable effects of a physiological O_2_ concentration (3%) on the amplification of erythroid progenitors implies a cooperation between HIFs and the Notch pathway. To this end, we utilized a model of early erythropoiesis ex vivo generated from cord blood CD34^+^ cells transduced with shHIF1α and shHIF2α at 3% O_2_ and 20% O_2_ in the presence or absence of the Notch pathway inhibitor. We observed that Notch signalization was activated by Notch2R–Jagged1 ligand interaction among progenitors. The inhibition of the Notch pathway provoked a modest reduction in erythroid cell expansion and promoted erythroid differentiation. ShHIF1α and particularly shHIF2α strongly impaired erythroid progenitors’ amplification and differentiation. Additionally, HIF/NOTCH signaling intersects at the level of multipotent progenitor erythroid commitment and amplification of BFU-E. In that, both HIFs contribute to the expression of Notch2R and Notch target gene *HES1*. Our study shows that HIF, particularly HIF2, has a determining role in the early erythroid development program, which includes Notch signaling.

## 1. Introduction

Multilevel action of physiologically low O_2_ concentration (<5%) as a regulatory factor in hematopoiesis/erythropoiesis has been well established in vivo and ex vivo [1,2]. Additionally, the adaptation of oxygenation to each stage of erythropoiesis by the level existing in vivo enhances the efficiency and production of RBC ex vivo [2]. Beyond the systemic regulatory feedback mechanism via the stimulation of mandatory erythropoietic hormone erythropoietin (EPO) production, a number of studies have shown that low O_2_ favors: (1) the production of EPO-unresponsive erythroid progenitors (BFU-E) from more primitive myeloid progenitors; (2) production of EPO-responsive erythroid progenitors (CFU-E), presumably from BFU-E; (3) CFU-E differentiation to generate an erythroid precursor [3,4,5,6,7,8,9]. In our previous work, we showed that this positive impact of low O_2_ concentrations on erythroid progenitors operates via shortening their cell cycle and, consequently, enhancing their amplification, but without impairing their proliferative capacity [2]. The aim of this study is to obtain insight into the analysis of possible underlying molecular pathways. Metabolic set-up in the response to a low O_2_ environment integrates a complex molecular network and in the first line is mediated by a family of transcription factors, hypoxia-inducible factors (HIFs) (predominately HIF-1 and HIF-2) [10]. They are heterodimeric proteins that comprise one of three isoforms of: the α subunit (HIF1-3α), stabilized when the O_2_ concentration is <5% or upon cytokine stimulation, and the β subunit, which is constitutively expressed [11,12]. Once stabilized, HIF-1α and HIF-2α are translocated in the nucleus where they are associated with β subunits in order to form functional HIF transcription factors that facilitate the transcription of a vast number of target genes, including those regulating erythropoiesis. In different cell models, HIFs interplay with other signaling pathways such as Notch at the level of the signal triggering, transcription or functional regulation [13,14,15]. This is an evolutionarily conserved signaling pathway that controls differentiation in most metazoan cell types [16]. Different models show the role of Notch signaling in hematopoiesis/erythropoiesis. In addition to its role in the maintenance and ex vivo expansion of human hematopoietic stem and progenitor cells [17,18], the activation of Notch in human and murine primitive hematopoietic cells (via the interactions with the Notch ligand expressing stroma or by the soluble ligand present in culture medium or the genetically constructed, constitutively active mutant) promotes their differentiation toward erythroid lineage [19,20,21,22].

The activation of Notch signalization is triggered by paracrine interaction of Notch receptors (Notch1-4) with specific ligands (Delta (DII 1,3,4) and Jagged/serrate (Jag1,2)) [23] expressed on a wide spectrum of hematopoietic and stromal cells. These include multipotent CD34^+^ progenitors as well as erythroid progenitors and early erythroblasts [24]. This implies that Notch signaling during hematopoiesis may be mediated not only by interactions between hematopoietic and stromal cells, but also by homotypic interactions between the hematopoietic progenitors themselves [25].

In this study, we purposely analyze the hypothesis that low O_2_-mediated stimulation of erythroid progenitor expansion comprises a crosstalk between Notch and HIF pathways. To that end, we utilized the protocol of ex vivo erythroid development from cord blood (CB) CD34^+^ cells with silenced HIF1α and HIF2α.

Our study showed that HIFs, particularly HIF2, have a determining role in the control of an early erythroid development program, which, among others, includes Notch signaling. We demonstrate that HIF/NOTCH signaling intersect at the state of multipotent progenitor erythroid commitment and the amplification of early erythroid progenitors, where HIF regulates Notch signaling concerning signal triggering and gene expression.

## 2. Materials and Methods

### 2.1. CD34^+^ Cell Isolation

Cord blood (CB) samples delivered (with the mother’s approval) to the Cell Therapy Unit of the French Blood Institute, Bordeaux, that had been rejected for banking were used for the experiments (in compliance with national French regulation, declared to the Ministry of Research: DC-2019-3720). CB CD34^+^ cells were isolated using an immunomagnetic technique (Miltenyi Biotec, Paris, France), giving a highly (at least 90%) enriched CD34^+^ fraction [26].

### 2.2. Generation of Lentiviral Vectors and Viral Particles

DNA constructs (21 bp sense and antisense oligonucleotides designed in the 3′-coding region of the human HIF-1α and HIF-2α genes) were kindly provided by the vectorology facility Vect’UB, University of Bordeaux, and were prepared as described in [27]. Briefly, DNA fragments (Eurogentec, Angers, France) were cloned in the pic20-plasmid behind the polymerase III H1 promoter. H1-shRNA sequences were subcloned in the lentivirus (pTripΔU3Ef1α-EGFPMCSΔU3), which contains the enhanced GFP (eGFP) gene under the control of the EF1αpromoter. Lentiviral supernatants were produced by transient CaCl2 transfection of HEK293T cells.

### 2.3. Transduction Protocol for CD34^+^ Cells

CB CD34^+^ cells (10^6^ cell/mL) were cultured for 72 h in Iscove medium(IMDM; Invitrogen, Cergy-Pontoise, France) supplemented with 15% bovine serum albumin (BSA)/insulin/transferrin (BIT 9500; Stemcell Technologies, Saint Égrève, France), recombinant human (rhu) stem cell factor (SCF; 100 ng/mL), Flt3-ligand (FL; 100 ng/mL),Interleukin-3 (IL-3; 60 ng/mL), and TPO (TPO; 20 nM). All the cytokines were purchased from Peprotech, Neuilly-Sur-Seine, France. Concentrated shRNA to HIF1α and shRNA to HIF2α were added at a concentration of 25 MOI (Multiplicity Of Infection) twice at 24 h intervals. The control was non-transduced cells (NT). After transduction, the cells were washed twice and counted, and sorted based on GFP expression, which was analyzed by fluorescence-activated cell sorting (FACS CANTO II, Becton Dickinson, Rungis, France). The results of transduction efficiency based on GFP expression in cells are shown in Figure 1.

### 2.4. Erythroid Cell Cultures

Transduced CDD34 cells with shRNA to HIF1 (shHIF1) α or to HIF2 α (shHIF2), as well as non-transduced cells (NT), were plated in the modified serum-free medium supplemented with 1% deionized bovine serum albumin, 120 ng/mL iron-saturated human transferrin, 900 ng/mL ferrous sulfate, 90 ng/mL ferric nitrate and 10 mg/mL insulin (Sigma-Aldrich, St Louis, MO, USA). In NT conditions, cells were cultivated in the presence or absence inhibitor of Notch signaling, the γ-Secretase inhibitor (Abcam, Cambridge, UK). Cultures were performed at O_2_ concentrations supposed to correspond to the physiological stages in erythroid development [2]. Using phase I (Day 0 to Day 8) from the protocol published by Douay’s group [28], 10^4^/mL cells were cultivated at 37 °C in a humidified atmosphere with 5% CO_2_/3% O_2_ (Proox Culture Chamber, Model C174 with the O_2_ regulator Pro-ox 110 and CO_2_ regulator Pro-CO_2_; Biospherix, Redifield, NY, USA) or 20% O_2_ (Incubator Igo 150 Cell Life; Jouan, St. Herblain, France) in the presence of 10^6^ M hydrocortisone (Sigma-Aldrich, St Louis, MO, USA), 100 ng/mL stem cell factor (SCF) (Peprotech, Neuilly-Sur-Seine, France), 5 ng/mL interleukin-3 (Peprotech), and 3 U/mL EPO (PBL Biomedical Laboratories, Piscataway, NJ, USA). The cells are incubated in presence or absence of Notch signaling pathway γ-secretase inhibitor DAPT (N-[N-(3,5-difluorophenacetyl)-l-alanyl]-s-phenylglycinet-butyl ester) (Abcam, Cambridge, UK) at the final concentration of 100 µM, which we previously determined as the concentration inducing a significant decrease in erythroid cell expansion without causing their apoptosis (data not shown). At selected time points, cells were stained with antibodies for flow cytometry or immunofluorescence, assayed for their CFC content, CFSE stained or proposed for real-time polymerase chain reaction (PCR), as described later.

### 2.5. CFC Assay

The hematopoietic committed progenitors (BFU-E, CFU-E, and CFU-granulocyte macrophage) were assayed as described previously [26]. Briefly, cultivated cells harvested at Day 4 or Day 8 were seeded in concentrations of 600/mL or 1200/mL, respectively, in the (a) cytokine-supplemented methycellulose Methocult 4034 (StemCell Technology, Saint Égrève, France), and colonies (BFU-E, CFU-granulocyte macrophage) were counted after 14 days of culture; (b) for the detection of CFU-E progenitors, the cultivated cells were seeded in only EPO-supplemented methycellulose Methocult 4330 (StemCell Technology, Saint Égrève, France) and the colonies were counted after 7 days of culture. Incubation proceeded at 37 °C in a humidified atmosphere with 20% O_2_ and 5% CO_2_.

### 2.6. Phenotypic Analysis

In order to characterize the differentiation level of erythroid cells during the culture, we analyzed the expression of glycophorin A (GPA) and transferrin receptor (CD71). GPA is absent from committed erythroid progenitors (i.e., BFU-E, CFU-E), but appears on morphologically recognizable basophilic erythroblasts and remains stably expressed on mature erythrocytes [29]. The transferrin receptor starts to be detected before GPA at the BFU-E stage, reaches a maximum on recognizable erythroblasts and progressively disappears at the late reticulocyte stage [29]. Additionally, in order to follow Notch receptor presentation during early erythropoiesis, Notch 1 and Notch2 receptor (R) expression was analyzed. Cells were harvested at different time points (from D-3 to D8) in phosphate-buffered saline with 1% human serum albumin. Cells were then stained with Brillant Violet 421-conjugated anti-CD71, phycoerythrin-conjugated anti-GPA (BD Biosciences, Le Pont-de-claix, France) and allophycocyanin-conjugated anti-Notch1R (Thermofisher scientific, Waltham, MA, USA) and phycoerythrin-conjugated anti-Notch 2R (Thermofisher scientific) for 25 min in dark and washed and analyzed on flow cytometer. Isotype control antibodies were used as controls for background fluorescence.

### 2.7. RNA Isolation and Quantitative Real-Time PCR

In order to examine Notch signaling expression during early erythropoiesis and its interplay with HIF-mediated pathways, cells were harvested at Day 4 and Day 8 of culture from all experimental conditions (NT, shHIF1 shHIIF2 at 3% and 20% O_2_) for real-time PCR analysis. The NOTCH target genes *HES1* and *HEY2* were assayed. For RNA isolation, 800 μL of TRIzol reagent (Invitrogen, Cergy-Pontoise, France) was added to the cells and RNA was prepared according to the manufacturer’s instructions. The reverse transcriptase (RT)/DNase step was performed with the QuantiTect Reverse Transcription kit (Qiagen, Courtaboeuf, France). Quantitative real-time PCR was performed using the Applied Biosystems StepOne Real-Time PCR System in a 25 μL reaction volume containing 5 μL of cDNA (1:20 dilution of the reverse-transcribed sample), 12.5 μL of SYBR Green SuperMix (Quanta Bioscience, Gaithersburg, MD, USA), and primer pairs at a final concentration of 0.5 μM. The PCR program included a denaturation step at 95 °C for 10 min, an amplification step for 40 cycles (15 s at 95 °C, 30 s at 60 °C) and a final dissociation curve step in order to determine the specificity of the product. Samples were duplicated for each run. Relative quantification was performed by calculating the 2^ΔΔCt^ value. Human β-actin, known to remain constant in the experimental conditions, was used as the reference gene to normalize all results [2,27]. Relative gene expression was calculated with respect to the control condition, cDNA obtained from CB CD34^+^ cells. The list of primers used for real-time quantitative PCR experiments is presented in the Appendix A.

### 2.8. CFSE Staining

Fluorescent dye CFSE was used for proliferative history analysis, as described previously [30]. Briefly, freshly isolated CD34^+^ cells in the presence or absence of DAPT were incubated for 30 min with CFSE dye (100 μM final) (Sigma-Aldrich, St Louis, MO, USA) according to the manufacturer’s instructions, washed extensively, and resuspended in cell culture medium. CFSE fluorescence of Day 0 CD34^+^ cells were measured on a flow cytometer (FACS CANTO II, Becton Dickinson, France) immediately after labeling and at the selected culture time points.

### 2.9. Propidium Iodide Cell Viability Assay

Assay is performed following previously published protocol based on the DNA intercalating agent propidium iodide (Beckman Coulter, Villepinte, France) with easy penetration in the damaged membranes of non-viable cells [31]. Briefly, 5 × 10^4^ cells harvested at Day 4 and Day 8 of culture were labeled with PI (10 mg/mL) for 15 min at 4 °C, washed in PBS buffer, and analyzed on a flow cytometer (Becton Dickinson Bioscience, FACS Canto II, Rungis, France). The intensity of the fluorescence was acquired in the yellow and red filter range.

### 2.10. Statistical Analysis

Data are expressed as the mean ± SD (standard deviation). When *n* < 30 and due to the absence of normal distribution, the experimental conditions were first tested with Kruskal–Wallis, then with the non-parametric Wilcoxon–Mann–Whitney paired comparison tests. Statistical significance is considered at *p* < 0.05.

## 3. Results

### 3.1. Erythroid Cell Expansion Is Abrogated When Notch and HIF-Associated Pathways Are Inhibited

We evaluated the erythroid cell expansion in the situation where the NOTCH or HIF-induced signaling pathways are inhibited, using the first phase of ex vivo erythropoiesis comprising the commitment of multipotent CD34^+^ hematopoietic progenitors to erythroid lineage, the apparition and amplification of erythroid progenitors and early precursor differentiation [2]. Low-O_2_-induced metabolic response is mediated majorly by HIF1 and HIF2 transcription factors. In order to elucidate their respective role, CB CD34^+^ cells were transduced with bicistronic lentiviral vectors carrying two distinct promoters, EF1a and H1, driving green fluorescent protein (GFP) (reporter) and a small hairpin RNA (shRNA) against the HIF1α or HIF2α subunits, respectively (sequences published by Rouault-Pierre et al. [27]). CD34^+^ CB was transduced (with 46% ± 11 for shHIF1α and 54% ± 14 for shHIF2α transduction efficiency) and FACS-sorted 3 days post-transduction, and cultured for an additional 8 days in the presence of growth factors at low (3%) and standard O_2_ concentrations (20%) (Figure 1a,b).

We observed that incubation at low O_2_ conditions alone markedly increased the total cell number compared with standard conditions, as expected. In addition, HIF2α silencing (shHIF2) and, to a lesser extent, HIF1α markedly reduced erythroid cell expansion in both O_2_ culture conditions (for shHIF2 > 90% of cell expansion reduction compared to NT condition at Day 8 (D8)) (Figure 2a,b). Regarding shHIF1α, these effects are significantly more pronounced at 3% O_2_ (approximately 80% inhibition in the shHIF1 condition compared to 60% in the same conditions at 20% O_2_). Thus, HIF stabilization (by environmental low O_2_ concentration or by growth factors present in the culture medium at 20% O_2_), particularly HIF2, exhibits a critical role in the early erythroid cell development ex vivo.

Considering the Notch juxtacrine pathway, it has been previously shown that it could be triggered by homotypic interactions between hematopoietic cells during erythroid development [25,32,33]. Taking this into account, we showed that the incubation of NT CD34^+^ cells in the presence of Notch signaling γ-secretase inhibitor DAPT significantly, but less pronouncedly than HIF, reduced the erythroid cell number at 3% as well as at 20% O_2_, affirming its role in the early erythroid development ex vivo. Additionally, the decrease in total cell expansion is associated with the decrease in cell survival in the case of the inhibition of HIF mediation, but not Notch signaling (Appendix A). That is the reason why we performed the tracking cell division history by CFSE staining. This showed that the inhibition of Notch signaling reduces the percentage of cells detected in the fraction of most divided cells. This indicates that the reduction in expansion in the presence of Notch inhibitor comes from the decrease in the proliferation rate of the early erythroid proliferative compartment (Appendix A).

### 3.2. Erythroid Progenitor Amplification Is Differently Modified When the HIF-Mediated or Notch Pathway Are Inhibited

In order to investigate if the observed cell expansion reduction is associated with the modulated erythroid progenitor’s amplifications, we assayed erythroid colony formation (Figure 2). Low oxygen culture alone induced a significant increase in erythroid progenitor number (total number of BFU-E and CFU-E is increased by two-fold at Day 4 and to four-fold at Day 8 comparing to standard O_2_ conditions) (Figure 3a,b). The analysis at Day 4 and Day 8 of culture showed that in shHIF1 and shHIF2 conditions, we detected significantly fewer early mature erythroid progenitors relative to NT 3% O_2_ (Figure 3a,b). While this effect is similar in shHIF2 conditions, the reduction in progenitors in the shHIF1 condition at 20% O_2_ is only slight (Figure 3a,b). In contrast, the inhibition of the Notch pathway results in only modest but significant diminution in BFU-E, as well as CFU-E progenitors at Day 4 and Day 8, no matter the oxygen culture concentration. At Day 8, while the number of BFU-E is still significantly reduced upon Notch inhibition, there are no differences in the CFU-E number with respect to NT conditions at both O_2_ concentrations. Furthermore, the Day 4 to Day 8 fold change in erythroid progenitor number revealed that inhibition of Notch alone did not impair the amplification of BFU-E progenitors relative to NT condition (20% O_2_ NT vs. NT + DAPT: 4 ± 1 vs. 4,3 ± 2; 3% O_2_ NT vs. NT + DAPT 6 ± 3 vs. 10 ± 4) (Figure 3c). Conversely, a significantly higher fold change is detected for CFU-E number (20% O_2_ NT vs. NT + DAPT: 19 ± 9 vs. 72 ± 30; 3% O_2_: NT vs. NT + DAPT: 61 ± 25 vs. 170 ± 40), which indicates that the inhibition of Notch stimulates differentiation of BFU-E toward CFU-E stage (Figure 3c). Additionally, silencing HIF1α and HIF2α significantly diminished the fold change in BFU-E number compared to the NT condition. However, in shHIF1, the same fold change in CFU-E number as detected for BFU-E reveals that, in this condition, BFU-E to CFU-E conversion is not impaired. This comes from the fact that newly formed CFU-E could be only those arisen from BFU-E progenitors, as CFU-E, unlike BFU-E, do not have self-renewing capacity [34]. In contrast, silencing of HIF2 completely abrogated the differentiation of BFU-E to CFU-E stage. Altogether, these results indicate the different and specific role of each exanimated signaling molecule in early erythropoiesis.

### 3.3. Erythroid Differentiation Is Altered When HIF-Mediated but Not Notch Pathways Are Inhibited

In order to characterize the differentiation level of erythroid cells during the culture, we analyzed the expression of glycophorin A (GPA) and transferrin receptor (CD71). GPA is absent from committed erythroid progenitors (i.e., BFU-E, CFU-E), but appears on morphologically recognizable basophilic erythroblasts and remains stably expressed on mature erythrocytes [29]. The transferrin receptor starts to be detected before GPA at the BFU-E stage, reaches a maximum on recognizable erythroblasts, and progressively disappears at the late reticulocyte stage [29]. We observed that at Day 4, CD71^+^/GPA^+^ precursor cells represent a minor fraction (Figure 4). Low oxygen alone (3% O_2_ NT) stimulates erythroid differentiation, according to a significant increase in percentage and absolute number of precursors cells (CD71^+^/GPA^+^) relative to 20% O_2_ (20% O_2_ NT). Additionally, inhibition of the Notch pathway increases the occurrence of GPA expression revealed as a significantly higher CD71^+^/GPA^+^ percentage of cells present at Day 4 as well as Day 8 of culture (Figure 4a, Appendix A). Thus, even though the frequency of precursor cells is higher in this condition, we did not obtain a higher total erythroblast number in culture with respect to the NT condition (Figure 4b). This was associated with a smaller number of detected erythroid progenitors, giving rise to the precursor cells that have been developed from CD34^+^ cells. In contrast, erythroid differentiation is severely impaired in shHIF1 and shHIF2 conditions relative to the NT condition, according to the percentage or total number of precursor cells at Day 4 and Day 8 of culture (Figure 4a,b). While the inhibition of Notch did not alter erythroid differentiation, it is severely impaired in the shHIF1 and shHIF2 conditions relative to the NT condition, according to the percentage or total number of precursor cells at Day 4 and Day 8 of culture (Figure 4a,b). These effects are observed at 3 as well 20% O_2_. Additionally, the analysis of overall erythroid commitment (progenitors and precursors cells) reveals that it is differently affected by Notch or HIF-mediated pathways (Appendix A). Thus, at Day 4, approximately 50% of cells belong to erythroid lineage (progenitors plus precursors) at 3% O_2_, and this number is 30% at 20% O_2_. At Day 8, all the cells are erythroid at both O_2_ concentrations. The inhibition of Notch alone does not change this development pattern, no matter the O_2_ culture concentration. In contrast, in shHIF1α and particularly shHIF2α conditions, approximately 90% of cells are non-erythroid lineage at Day 8, revealing severe inhibition of the erythroid differentiation from CB CD34^+^. In this situation, when erythroid progenitor development from multipotent CD34^+^ cells as well as the amplification of progenitors and their differentiation toward precursors are blocked, detected Day 4 to Day 8 cell expansion is related to non-erythroid cells. This suggests that the effects of silencing HIFs, particularly shHIF2α, are selective for early erythroid commitment stages.

### 3.4. Notch Signaling Is Modified When HIF1α and HIF2α Are Silenced

In order to investigate Notch/HIF interplay in early erythroid development, we started our exploration at the level of signal triggering and analyzed the Notch receptor 1 and 2 (Notch1R, 2R) expression. Flow cytometry analysis revealed that low levels of both receptors are present in native freshly isolated or thawed CD34^+^ cells (Figure 5a). During the erythroid culture, the expression of Notch1R remained at the similar minimal level without significant differences between experimental conditions (data not shown). In contrast, erythroid development comprises a progressive significant increase in the percentage of Notch2R-expressing cells, as well as its quantity per cell (Figure 5a,b), resulting in the presence of Notch2R in almost all the cells by Day 8.

Silencing of both HIF1 and HIF diminished the expression of Notch2R on Day 4. At the erythroid stage of development corresponding to Day 4 to Day 8, the inhibition of Notch2R remains markedly inhibited, which is particularly pronounced in the shHIF1 condition (Figure 5a). These phenomena were detected in both O_2_ culture conditions. This indicates that stabilized HIF1 and HIF2 enhance the expression of Notch2R. Additionally, keeping in mind that most D8 cells in the shHIF1 and shHIF2 condition are non-erythroid, a decrease in the Notch2R expression with respect to the NT condition could be the consequence of the low Notch2R expression detected during the development of other hematopoietic lineage. Furthermore, juxtacrine Notch signaling in our system is mediated by the interaction between the Notch2R and Jagged1 ligand, which was the only Notch ligand that we could detect (Appendix A). Finally, in order to determine whether Notch–HIF interplay has the effect at the level of gene expression, we estimated the expression of the Notch target *HES1* and *HEY2* genes in native CD34^+^ and erythroid cells at Day 4. We did not detect the expression of *HEY2* in early erythropoiesis. In contrast, the expression of *HES1* was significantly reduced in the shHIF1 and shHIF2 conditions, indicating that there is Notch/HIF interplay at the transcription level (Figure 5c).

## 4. Discussion

Our study revealed that concomitant activation of Notch and HIF-mediated path-ways, which interplay at the level of the gene expression as well as signal activation, plays a critical role in early erythropoiesis ex vivo. Our 8-day culture system of early erythropoiesis comprises several developmental stages: expansion of multipotent progenitors CD34^+^, their differentiation toward erythroid progenitors, the apparition of early erythroid progenitors self-renewing BFU-E (Day 0–Day 4), and their conversion to mature erythroid progenitors CFU-E and early precursors (Day 4–Day 8°).

Hence, this model gives a suitable system to investigate different erythroid regulatory factors.

As expected and in agreement with our previous results, as well as the results of other authors, a culture maintained at a physiological low O_2_ concentration (1.5–5%) favors the production of erythroid progenitors (BFUE and CFU-E), and their differentiation into early erythroid precursors [2,3,4,5,6,8,9]. The impact of O_2_ could be different regarding oxygen content. Thus, at 1% O_2_, the production and amplification of BFU-E seem to be enhanced, while their commitment toward CFU-E and further is accomplished at higher O_2_ concentrations [7]. Higher O_2_ concentrations (3–5%) are needed for the expansion and differentiation of committed myeloid progenitors (colony-forming cells) [26]

In order to distinguish the role of HIFs as a major cellular molecular mediator in a low O_2_ environment in these processes, we created mutant CD34^+^ cells silenced for HIF1α and HIF2α. We observed that HIFs, and particularly HIF2, are indispensable for the commitment of the multipotent hematopoietic progenitors to primitive erythroid lineage cells and their further expansion and differentiation (Figure 1, Figure 2, Figure 3 and Figure 4). Additionally, we observed that BFU-Es obtained in culture propagated from CD34^+^ cells silenced for HIf1 and HIF2 are capable of multiplying. However, while in the shHIF1 condition, BFU-E are capable of converting to the CFU-E stage, and in the shHIF2 condition, the maturation is blocked at the BFU-E stage. This indicates that HIF1 and HIF2 have specific and complementary roles in early erythropoiesis.

Our findings are concordant with others showing the predominant role of HIF2 in adult hematopoiesis [35,36,37,38,39]. Thus, it was shown that the knockdown of HIF2α and, to a much lesser extent, HIF-1α impedes the long-term repopulating ability of human CB CD34^+^ cells and significantly diminishes their capacity to form erythroid colonies [27].

In contrast, HIF-mediated regulation comprises the regulation with other signaling pathways [40]. In addition to its role in the ex vivo expansion of human hematopoietic stem and progenitor cells [17,18,33], the activation of Notch signaling promotes their differentiation toward erythroid lineage [19,21,41]. Additionally, in vivo model using tamoxifen-inducible CreER knock-in mice for individual Notch receptors, in combination with a Notch reporter strain (Hes1GFP), showed the critical role of the Notch pathway in early hematopoiesis and in the commitment of hematopoietic progenitors toward erythroid lineage [24].

In our system, we used the protocol of ex vivo early erythropoiesis to show that juxtacrine Notch is activated due to homotypic interactions between Notch receptors and the Jagged-1 ligand in differentiating erythroid cells. In our model, Notch1 and Notch2 receptors are differently expressed; when both are present in multipotent hematopoietic progenitors, Notch2R expression increased, suggesting its dominant role in early erythroid development. This is coherent with other studies showing that its expression peaks in parallel with CFU-E and pro-erythroblast production and begins to decrease at the stage of basophilic erythroblasts [32,33].

In our model, Notch inhibition resulted in a slight but significant reduction in erythroid cell expansion, which was not associated with higher apoptosis but only with a diminution of cell division (Figure 2, Appendix A). Additionally, Notch inhibition resulted in approximatively 50% ± 15 of reduction in total cell number at 20% O_2_ vs. approximately 30% ± 13 detected at the same day point at 3% O_2_. This is in agreement with the increase in the erythroid cell expansion and differentiation, detected at a physiological O_2_ concentration with respect to 20% O_2_ [2]. The reasons for this are associated with both the increased stabilization of HIF factors at a low O_2_ concentration that exhibits favorable effects in the early erythropoiesis, and inversely a reduction in erythroid development provoked by hyperoxygenation at 20% O_2_ [1,2]. The favorable effects of physiological O_2_ concentration yield cells more resistant against Notch inhibition.

In addition, even the amplification of early erythroid progenitors was diminished, and the inhibition of Notch did not impair erythroid progenitor development from CD34^+^ multipotent progenitors and their development to precursors. This implicates that Notch activation in our model with early erythropoiesis ex vivo affects mostly the maintenance and amplification of the multipotent progenitors’ (MPP, CMP, MEP) compartment and their commitment toward the erythroid lineage progenitor. Furthermore, we noticed that the inhibition of the Notch pathway even promotes BFU-E to CFU-E commitment as well as increases the percentage of more mature GPA^+^ erythroid cells (Figure 3c, Appendix A). Altogether, the effects of Notch inhibition reflect the implication in erythroid cell proliferation and delay the differentiation, as previously shown [33,42].

Ligand binding to the Notch receptor induces the receptor cleavage by ADAM metalloprotease, which forms Notch extracellular truncation. Then, the final cleavage is mediated by y secretase, resulting in the release of the Notch intracellular domain (NotchICD) that translocates in the nucleus. It forms a trimeric complex with the transcriptional regulator protein CBF1–Suppressor of Hairless–LAG1 and the co-activator mastermind-like transcriptional co-activator1 in humans and stimulates the transcription of target genes [43].

Numerous experimental systems have revealed that the Notch/HIF interplay could be realized at the transcription level: HIF-mediated stabilization of NotchICD stabilization induces the expression of Notch target genes, and the signal triggering level: HIF induces the expression of Notch receptor and ligands [13]. Our results fit these findings: we observed that HIF1α as well as HIF2α are implicated in the Notch2R expression. In addition, NOTCH2ICD stabilization is abrogated when HIF1α and HIF2α are silenced, resulting in the reduction in the Hes target gene expression. Importantly, in our model, the inhibition of Notch/HIF interplay impairs the early erythroid development, suggesting its determining role in it.

It should be stressed, however, that we detected pronounced erythroid differentiation inhibition when HIF1α and HIF2α were silenced, giving rise to most of the expanded cells being non-erythroid at Day 8 (Appendix A). Considering this decrease in the Notch2R expression in respect to NT condition could be the consequence of a rise in the expression of Notch2R associated with the development of other hematopoietic lineage (e.g., megakaryocyte progenitors) [24].

It should be noted that Notch signaling could potentiate HIF-mediated responses by acting as a molecular sink for FIH, or by favoring the expression of hypoxia-induced genes [44,45]. This should be taken into account when Notch/HIF interplay is considered.

We showed here that similar phenomena observed at 3%O_2_ were detected at 20% O_2_. This could be associated with the action of “hypoxia mimicking agents” (i.e., SCF) presented in the culture medium, which stabilize HIF even at a high O_2_ [11]. However, the significant advantage in the erythroid expansion observed at 3% O_2_ suggests that above Notch/HIF interplay, it comprises other signaling pathway implications or/and it is associated with erythroid cell oxidative status [2].

Altogether, our results give light to the early erythropoiesis regulatory network, giving way for the optimization of the protocol for the ex vivo erythropoiesis. The utilization of Notch ligands as well as HIF stabilizers (hypoxia mimicking agent) could be useful for improving the ex vivo erythroid culture

## Figures and Tables

**Figure 1 biomolecules-12-00540-f001:**
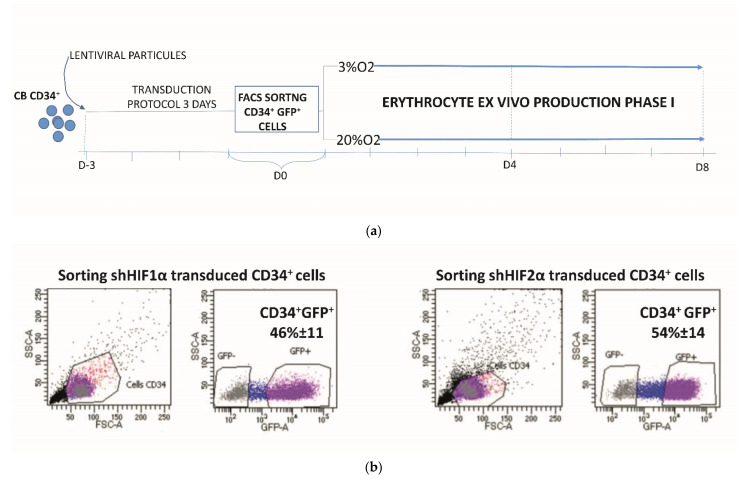
Experimental design. (**a**) Cord blood CD34^+^ cells (10^6^ cell/mL) were cultured (Day 3 to Day 0) for three days and transduced with shRNA to HIF1α and shRNA to HIF2α at a concentration of 25 MOI (Multiplicity Of Infection) twice at 24 h intervals. As control, we used non-transduced cells (NT). Then, transduced cells were sorted based on Green Fluorescence Protein (GFP) expression by FACS and plated in the erythroid culture protocol for 8 days (Day 0 to Day 8). Non-transduced CD34^+^ cells in presence or absence of Notch γ-secretase inhibitor DAPT as well as transduced CD34^+^ cells with shHIF1α or with shHIF2α were grown in erythroid medium for 8 days at 20% and at 3% O_2_. At selected time points, cells were stained with antibodies for flow cytometry or immunofluorescence, assayed for their CFC content, underwent CFSE staining or were proposed for real-time polymerase chain reaction (PCR) as described in materials and methods. (**b**) The representative dot plot showing FACS sorting based on GFP expression giving the transduction efficiency of lentiviral particles containing shHIF1 and shHIF2 in CD34^+^ cells.

**Figure 2 biomolecules-12-00540-f002:**
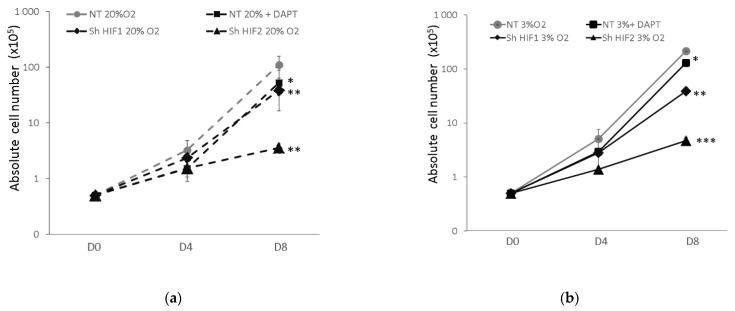
Erythroid cell expansion Non-transduced CD34^+^ cells in presence (■) or absence (●) of Notch γ-secretase inhibitor DAPT as well as transduced CD34^+^ cells with shHIF1α (⯁) or with shHIF2α (▲) were grown in erythroid medium for 8 days at 20% (dashed line, (**a**)) and at 3% O_2_ (solid line (**b**)). Data represent the absolute number of cells obtained from 50 × 10^3^ CD34^+^ cell at Day 0. The values are expressed as the mean ± SD of seven independent experiments at the logarithmic scale. Asterisks indicate a significant difference with respect to NT condition, at *p* < 0.05 (*), *p* < 0.01 (**), *p* < 0.001 (***), Mann–Whitney test. NT, non-transduced CD34^+^ cells; D, day of culture.

**Figure 3 biomolecules-12-00540-f003:**
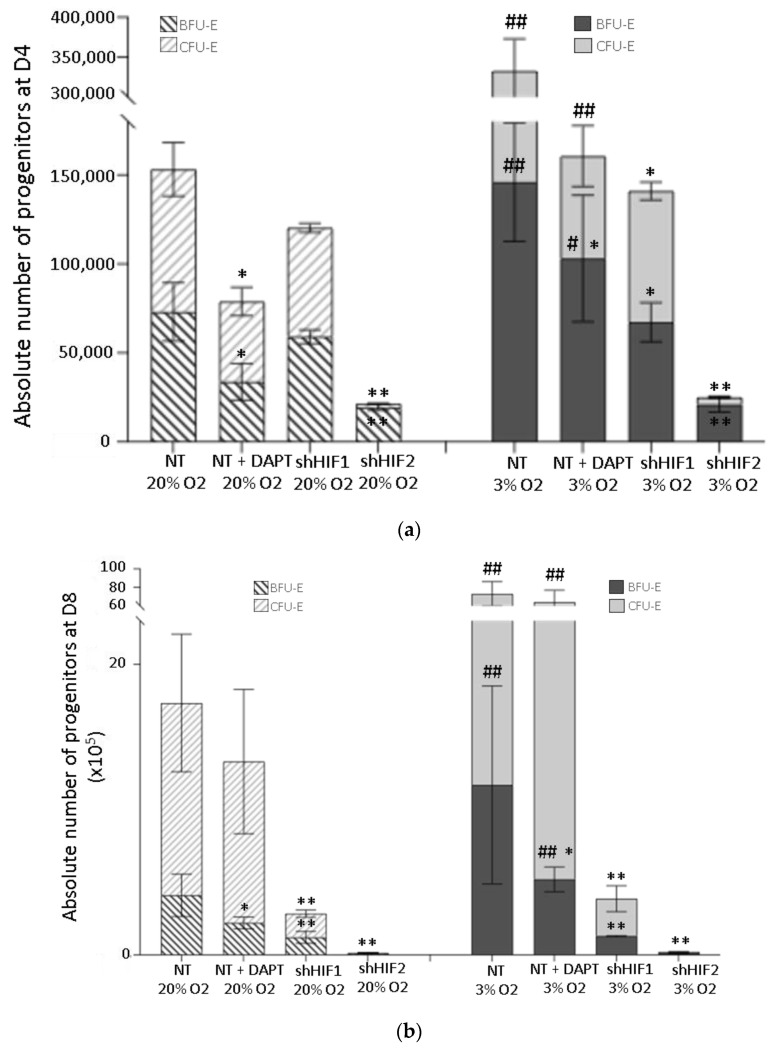
Production of erythroid progenitor. Non-transduced CD34^+^ cells in presence or absence of Notch γ-secretase inhibitor DAPT as well as transduced CD34^+^ cells with shHIF1α or with shHIF2α were grown in erythroid medium for 8 days at 20% and at 3% O_2_. The numbers of burst-forming unit erythroids (BFU-E) and colony-forming unit erythroids (CFU-E) were estimated after 4 (**a**) and 8 days of culture (**b**). Bars represent average values of total number of BFU-E and CFU-E obtained at Day 4 and Day 8 of culture. (**c**) The values show average fold change in BFU-E or CFU-E Day 8 number relative to Day 4. Data are presented as mean ± SD of seven independent experiments. Asterisks indicate a significant difference with respect to NT condition, at *p* < 0.05 (*), *p* < 0.01 (**); hashtags indicate a significant difference with respect to 20% O_2_ condition, *p* < 0.05 (#), *p* < 0.01 (##), Mann–Whitney test. NT, non-transduced CD34^+^ cells; D, day of culture.

**Figure 4 biomolecules-12-00540-f004:**
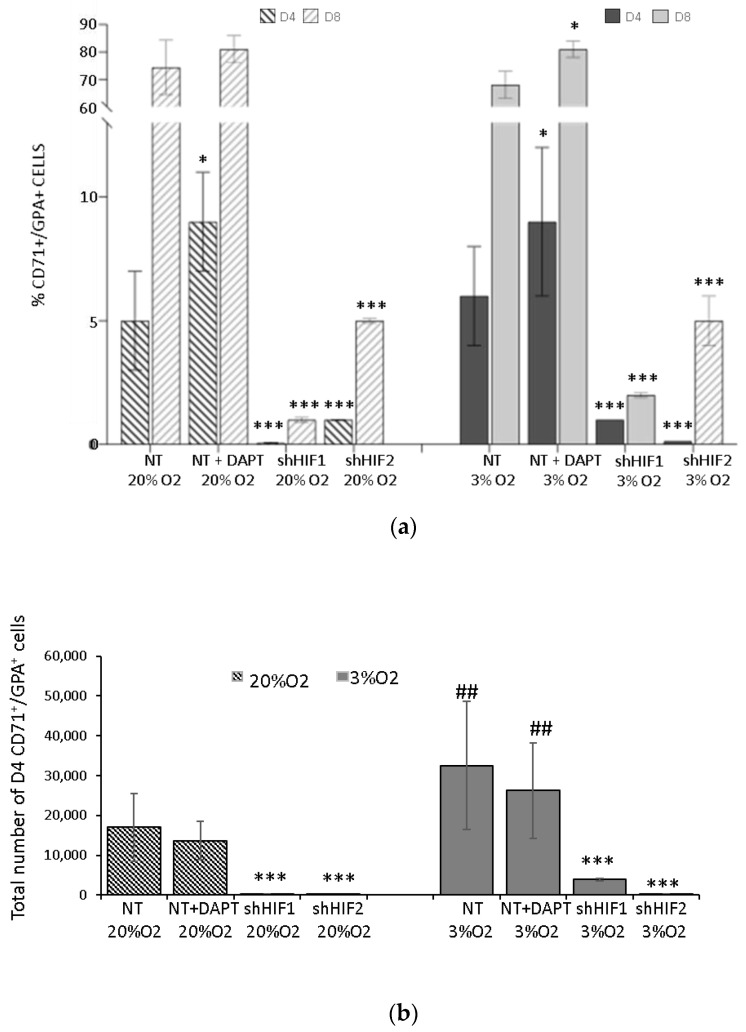
Erythroid differentiation. (**a**) Glycophorin A (GPA) and Transferrin receptor expression on erythroid cells obtained at Day 4 and Day 8 of culture of non-transduced CD34^+^ cells in presence or absence of Notch γ-secretase inhibitor DAPT as well as transduced CD34^+^ cells with shHIF1α or with shHIF2α at 20 and 3% O_2_. The bars show the average percentage of GPA^+^/CD71^+^ cells. (**b**) The average value of total number of GPA^+^/CD71^+^ cells in 4-day culture. Data are presented as mean ± SD of seven independent experiments. The asterisks indicate a significant difference with respect to NT condition at *p* < 0.05 (*), *p* < 0.001 (***); the hashtags indicate a significant difference with respect to 20% O_2_ condition at *p* < 0.01 (##), Mann–Whitney test. NT, non-transduced CD34^+^ cells; D, day of culture.

**Figure 5 biomolecules-12-00540-f005:**
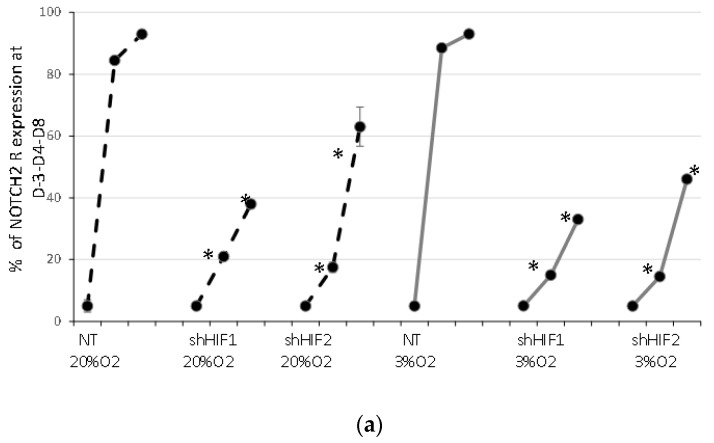
Notch/HIF interplay in early erythroid development. (**a**) The expression of Notch2 receptors in the cells harvested at the beginning (Day 3, D-3), Day 4 (D 4) and Day 8 (D 8) of eight-day erythroid cultures. The curves represent the average percentage of receptor expression in whole culture. (**b**) The bars show average mean fluorescence intensity. Data are presented as mean values ± SD of 4 independent experiments. (**c**) The expression of Notch target gene *HES1* was evaluated in erythroid cells obtained at Day 4 of culture of non-transduced CD34^+^ cells as well as transduced CD34^+^ cells with shHIF1α or with shHIF2α at 20 and 3% O_2_ by real-time polymerase chain reaction analysis. Bars represent relative gene expression with respect to native CB CD34^+^ cells. Data are shown as mean ± SD of five independent experiments. The asterisks indicate a significant difference with respect to NT condition at *p* < 0.05 (*), *p* < 0.001 (***), Mann–Whitney test. NT, non-transduced CD34^+^ cells; D, day of culture.

## Data Availability

All data are conserved in the internal server of EFS-NVAQ Bordeaux and are available upon request.

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
