# Peer review of "Regulatory Crosstalk between Physiological Low O2 Concentration and Notch Pathway in Early Erythropoiesis"

_biomolecules, 2022, doi:10.3390/biom12040540_

Round 1

Reviewer 1 Report

The study “Regulatory crosstalk between physiological low O2 concentration and Notch pathway in the early erythropoiesis” by V. Labat et al, is an interesting one. The authors have tried to explore the signalling integration between hypoxia-inducible factors (HIF) and Notch signalling in erythropoiesis, specifically in the early stages of erythroid cell differentiation. The authors have used cord blood (CB) cells to induce erythroid differentiation at two different O2 concentration to investigate how these two signalling pathways influence early erythroid cell generation from CD34+ CB cells. The model system is quite good and the aim of the study to dissect the collaborative roles of HIFs and Notch factors in early erythropoiesis is a rational one.

The study has some interesting findings such as the strong inhibition of proliferation and differentiation of early erythroid cells by silencing HIF1a and HIF2a expression in the CB cells. However, the major problem of the study is that it has not proved that HIF and Notch signalling collaborate to control the events regulating early erythroid cell differentiation. Although, the data provided shows a strong effect of HIF silencing on early erythropoiesis, the data for Notch signalling are rather weak and not convincing. The data provided to link HIF and Notch signalling, such as the reduced Notch2 expression, and absence of Notch intracellular domain (NICD) in the nucleus in the situation of HIF silencing are rather correlative. Absence of a strong evidence to show that these two signalling pathways interact to regulate early erythropoiesis undermines the significance of this study. My concerns are as mentioned below. Addressing those will significantly enhance the quality of the manuscript and make the findings relevant.

Major concerns:

  1. Overall, the data provided put Notch2 signalling downstream to HIF activity. However, whereas HIF silencing shows a strong effect on proliferation and differentiation, inhibition of Notch signalling does not show a similar effect in any aspects. Why? This suggests that HIF signalling independently controls events that are critical for early erythropoiesis and its link with Notch signalling is rather minimal. This needs clarification.
  2. In the abstract, the authors concluded, “Importantly, in our model, inhibition of Notch/HIF interplay impaired the early erythroid development, suggesting its determining role in it”. However, where is the proof that their interaction led to the observed effects in erythropoiesis?
  3. Concerning Fig. S1, the effect of Notch inhibition on cell division/cell cycle is not really great as most of the cells are in the same number of division as in the control. This suggests either Notch signalling was not affected by the inhibitor or Notch does not really play a role in cell cycle here. It is not clear from the data provided.
  4. It is not mentioned what DAPT concentration was used in the study to inhibit Notch activity.
  5. In Fig. 2, Notch inhibition by DAPT is not really strong on cell proliferation in 20% O2. But at 3% O2, it is nearly 50% inhibited. Why this discrepancy? Is Notch signalling redundant or not involved in cell proliferation at 20% O2?
  6. In Fig. 3, it is apparent that Notch inhibition has minimal effect in inhibiting CFU-E proliferation at both D4 and D8 at both O2 Does that mean CFU-E proliferation is independent of Notch signalling?
  7. From Fig. 4 it is apparent that Notch inhibition has no effect on differentiation at any time point and at any O2 This should suggest Notch has no role in differentiation and importantly will question whether Notch has any role in erythroid progenitor proliferation? Because as shown in earlier sections that Notch inhibition reduced proliferation and cell cycle, then it is difficult to understand how the total number of differentiated cells remain unaffected to Notch inhibition. However, the effects of HIF1 & 2 inhibition still hold, as the total number of cells are significantly less as has been shown in Figs. 1-3.
  8. What about cell survival in HIF1 & 2 inhibition? Do the cells die? No data have been provided on this aspect. If proliferation, cell cycle and differentiation are strongly affected to HIF inhibition, the cell death should also be affected.
  9. Concerning lines 348-352, the effect of HIF inhibition was more prominent in case of HIF2 as shown in all the Figures until now. However, as shown in Fig. 5a, the Notch2R expression is strongly reduced in case of HIF1 inhibition. HIF2 inhibition has still significant Notch2R expression at both O2 So, how does this observation reconcile with the strong inhibition in proliferation for HIF2 inhibition?
  10. In Fig. 5c, the data provided are not very convincing about HIF silencing-induced alteration in nuclear NICD levels. Please show more cells and a quantification of nuclear NICD will be more appropriate. But anyway, it is surprising to see the nuclear NICD is so strongly diminished against HIF silencing as in this situation Notch signalling is not completely abolished rather it is reduced as is shown in Fig. 5a &b.
  11. In Fig. 5d, again it is surprising that in case of HIF2 inhibition, HES expression is the minimum compared to that for HIF1 inhibition. Fig. 5a & b show that Notch2R is less expressed in case of HIF1 inhibition; hence, HES expression should be most affected there rather than for HIF2 inhibition.

Minor concerns:

  1. Line 50 should be ….vast number of target genes…..
  2. Line 52 should be ….is an evolutionarily conserved……
  3. Line 71 should be …crosstalk between Notch and HIF……
  4. Line 216, …..and to a lesser extent…..
  5. Figure 2 legend: fill in the brackets appropriately.
  6. Line 362, should be HEY2.

Author Response

Major concerns:

  1. Overall, the data provided put Notch2 signalling downstream to HIF activity. However, whereas HIF silencing shows a strong effect on proliferation and differentiation, inhibition of Notch signalling does not show a similar effect in any aspects. Why? This suggests that HIF signalling independently controls events that are critical for early erythropoiesis and its link with Notch signalling is rather minimal. This needs clarification.

We agree with the reviewer remarks. We added three additional experiments and performed  detailed statistic analysis. According to these, the significant differences were modified at the figure 3 and 4 in new R1 version. Based on all rectified data and considering reviewer’s comments we could now conclude that:

Silencing of HIF1α particularly HIF 2α induces strong block in early erythroid development manifested as inhibition of multipotent CD34+ progenitor commitment to erythroid lineage, apparition of  early erythroid self-renewing progenitor BFU-E, conversion of BFU-E to mature CFU-E and their differentiation toward first  erythroid precursors. On the other hand, inhibition of Notch induces modest but significant reduction of erythroid cell expansion, which was not associated with a higher apoptosis respect to NT condition (Figure 2, Figure S1). In addition, even the amplification of BFU-E was diminished, inhibition of Notch did not impair erythroid progenitor development from CD34+ multipotent progenitors and their development to precursors (Figure 3 and Figure 4).  Furthermore, we noticed that the inhibition of Notch pathway even promotes BFU-E to CFU-E commitment as well increases the percentage of more mature GPA erythroid cells .These  effects reflect the implication of Notch signaling in the multipotent CD34+ progenitor commitment toward erythroid lineage,  the amplification of BFU-E , and delay the differentiation. Also, these results reveal that Notch signaling does not have a critical, but rather a regulatory contributing role in early erythropoiesis ex vivo. In addition, considering the effects of both pathways, their roles are superposed at level of erythroid commitment and production of erythroid progenitor BFU-E (Day 0 to Day 4 of erythropoiesis ex vivo). Also, we demonstrated that this interplay is mediated at the level of signal triggering (Notch2 R expression) and gene expression (HES1 gene expression) (Figure 5, and answers to comments 9 and 11). All these issue were now clearly presented (results section) and minutely discussed (discussion section)

  1. In the abstract, the authors concluded, “Importantly, in our model, inhibition of Notch/HIF interplay impaired the early erythroid development, suggesting its determining role in it”. However, where is the proof that their interaction led to the observed effects in erythropoiesis?

The abstract is now modified following the conclusions mentioned in the answer 1.

  1. Concerning Fig. S1, the effect of Notch inhibition on cell division/cell cycle is not really great as most of the cells are in the same number of division as in the control. This suggests either Notch signalling was not affected by the inhibitor or Notch does not really play a role in cell cycle here. It is not clear from the data provided.

We presented CFSE staining in order to show the proliferation rate of culture cells. CFSE has covalent binds inside and its fluorescent intensity is diminished by every cell division. While measuring the fluorescent intensity in respect to D0 CFSE labelled cells, we can track cell division history in the culture (PMID: 20671746).  In our model, the inhibition of Notch signaling is inhibited, there is one modest but significant fraction of cells that divided less than the majority, giving less progeny cells. This could be associated with the decrease in total cells number in respect to NT condition since there was no difference in the cell survival between these two conditions (Figure S1). Also, considering that the Notch signaling is implicated in the maintenance of the multipotent progenitors with high proliferative capacity (PMID: 34450041), the observed results with CFSE staining could be related with the fact that Notch inhibition could provoke their reduction and development into cells/ progenitors exhibiting a decreased amplification rate. This is now more precisely explain (page 5, line 239-243)

  1. It is not mentioned what DAPT concentration was used in the study to inhibit Notch activity.

           DAPT concentration used in the experiments is now mentioned in paragraph 2.4.

  1. In Fig. 2, Notch inhibition by DAPT is not really strong on cell proliferation in 20% O2. But at 3% O2, it is nearly 50% inhibited. Why this discrepancy? Is Notch signalling redundant or not involved in cell proliferation at 20% O2?

 At the Figure 2, total cell expansion showed that at Day 8 of culture, the inhibition of Notch resulted in approximatively 50%±15 of reduction in total cells number vs approximately 30%±13 detected at the same day point at 3% of O2. This is in according with the increase of the erythroid cells expansion and differentiation, detected at physiological O2 concentration in respect to 20%O2. The reasons for this are associated with both increased stabilization of HIF factors at low O2 concentration that exhibits favorable effects in the early erythropoiesis (Vlaski et al 2009, PMID: 1937564; Vlaski-Lafarge &Ivanovic, 2015, PMID: 2652720), and inversely a reduction in erythroid development provoked by hyperoxygenation at 20%O2. The favorable effects of physiological O2 concentration yield cells more resistant against Notch inhibition (Figure 2).  These points are now mentioned in the discussion section (page 13, lines 510-520)

  1. In Fig. 3, it is apparent that Notch inhibition has minimal effect in inhibiting CFU-E proliferation at both D4 and D8 at both O2 Does that mean CFU-E proliferation is independent of Notch signalling?
  2. From Fig. 4 it is apparent that Notch inhibition has no effect on differentiation at any time point and at any O2 This should suggest Notch has no role in differentiation and importantly will question whether Notch has any role in erythroid progenitor proliferation? Because as shown in earlier sections that Notch inhibition reduced proliferation and cell cycle, then it is difficult to understand how the total number of differentiated cells remain unaffected to Notch inhibition. However, the effects of HIF1 & 2 inhibition still hold, as the total number of cells are significantly less as has been shown in Figs. 1-3.

Answers to 6 and 7: As it was mentioned in the answer 1, our rectified data presented at Figure 3 and Figure 4 in R  version as well as Figure S3a showed that Notch promotes conversion of BFU-E to CFU-E stage as well as it increases the percentage of more mature GPA+ erythroid precursors cells. These effects reflect the implication of Notch signaling (maintenance multipotent progenitors, their commitment toward erythroid lineage, amplification of BFU-E and delayed in the differentiation. These results are concordant to those previously showed implication of Notch signaling in undifferentiated state maintenance. (PMID 3445004).

  1. What about cell survival in HIF1 & 2 inhibition? Do the cells die? No data have been provided on this aspect. If proliferation, cell cycle and differentiation are strongly affected to HIF inhibition, the cell death should also be affected.

We estimated the cell survival with propidium iodide staining*(PMID: 27371595). Detailed procedure is explained in the paragraph 2.9 in R1 version. The percentage of  dead cells in all the experimental conditions is now presented at the Figure S1 as mean of percentage of PI positive cells of three independent experiments.

*Commonly we did that with Annexin V PI assay but in our kit, Annexin V is conjugated with FITC which is incompatible with GFP expressed in the transduced cells. Thus we could not use it.  Second issue, the new order of the Annexin V kit  with the compatible fluorochrome and following  new experiments would take significantly more time than what was accorded for the submission of revised manuscript.

  1. Concerning lines 348-352, the effect of HIF inhibition was more prominent in case of HIF2 as shown in all the Figures until now. However, as shown in Fig. 5a, the Notch2R expression is strongly reduced in case of HIF1 inhibition. HIF2 inhibition has still significant Notch2R expression at both O2 So, how does this observation reconcile with the strong inhibition in proliferation for HIF2 inhibition?

Answers to 9 and 11 : In order to evaluate interplay between Notch and HIF, the analysis of Notch2R expression showed that inhibition of HIF1 and HIF 2 significantly diminished its expression at Day 4 vs NT cells indicating that HIF1α as well as HIF2α are implicated in the Notch2R expression. (Figure 5a). It was shown by others that low O2 induces Notch2R expression. Also, at the same culture point, the expression of Notch target gene HES1 was significantly reduced in shHIF1 and shHIF2 condition vs NT condition, indicating Notch signaling/ HIF pathway interplay at the transcription level also (Figure 5c in R1 version). The difference which was observed  between shHIF1 and  shHIF2 condition in respect of HES1 expression, even the level of Notch2R was similar in both conditions, which could indicate that while both HIF are implicated in Notch2R expression, in our model, HIF2 acts  particularly at the level of gene  expression according to what was previously shown (PMID: 20678473).

However, from Day 4 to Day 8, Notch2R expression is rising albeit HIF inhibition. Keeping in the mind that at Day 8 the most of the cells are non-erythroid, the observed rise of Notch2R could reflect its expression during development of other hematopoietic lineage  (e.g; megakacocyte progenitors) (PMID: 23791481) and in which Notch2 expression is less depending on HIFs particularly HIF2.

All these issue were now clearly presented (results section, paragraph 3.4) and minutely discussed (discussion section, lines 546-551)

  1. In Fig. 5c, the data provided are not very convincing about HIF silencing-induced alteration in nuclear NICD levels. Please show more cells and a quantification of nuclear NICD will be more appropriate. But anyway, it is surprising to see the nuclear NICD is so strongly diminished against HIF silencing as in this situation Notch signalling is not completely abolished rather it is reduced as is shown in Fig. 5a &b.

Due to submission time shortage, in the absence of new experiments giving confirmed and convincing data, we decided not to present immunofluorescence experiments in R1 version.

  1. In Fig. 5d, again it is surprising that in case of HIF2 inhibition, HES expression is the minimum compared to that for HIF1 inhibition. Fig. 5a & b show that Notch2R is less expressed in case of HIF1 inhibition; hence, HES expression should be most affected there rather than for HIF2 inhibition.

Please  refer  answer  to comment  9.

  Minor concerns:

  1. Line 50 should be ….vast number of target genes…..
  2. Line 52 should be ….is an evolutionarily conserved……
  3. Line 71 should be …crosstalk between Notch and HIF……
  4. Line 216, …..and to a lesser extent…..
  5. Figure 2 legend: fill in the brackets appropriately.
  6. Line 362, should be HEY2.

All the minor concerns are fulfilled in the R1 version.

Reviewer 2 Report

In this study, the authors evaluated the crosstalk between the Notch pathway and the HIF1/HIF2 pathway.
Interestingly the inhibition of Notch/HIF impaired erythroid development.
These data were significant and informative.
However, I raised somepoints to improve the content of the report.
1. p2 lane 51. Quotation is missing eg PMID: 25809665 PMID: 11313887
2. p4 lane 153. blank character should be inserted
3. Human b-actin was used as a house keeping gene and claimed to be constant in
the experimental conditions. Can the authors give the referende source. Otherwise
a second house keeping gene like L28 should be introduced
4. Paragraph 2.9: Please provide detailed informations about the antibodies used
in the experiments
5. results section: The authors used 3% o2. The impact of the o2 concentration might
be different in regard to the oxygen content. Do the others have data of 1% O2 experiments?
6. Figure 2: the figure legend is partialy incomplete. Please add informations inside brackets

Author Response

  1. p2 lane 51. Quotation is missing eg PMID: 25809665 PMID: 11313887

Quotations PMID 25809665 and PMID 11313887 were  added in the R1 version of the manuscript.

  1. p4 lane 153. blank character should be inserted

The blank characters were checked through the whole manuscript.

  1. Human b-actin was used as a house keeping gene and claimed to be constant in

the experimental conditions. Can the authors give the referende source. Otherwise

a second house keeping gene like L28 should be introduced.

In our previous work (Vlaski et al 2009, PMID: 19413727), we tested β-actin expression in function of different O2 culture concentrations and we confirmed that its expression is constant no matter the    level of culture oxygenation. In addition, the silencing of HIF1α and HIF2α a in CD34+ cells does not change the expression of β-actin (PMID: 24095676).These references are used in the R1 version in order to  claim that β-actin expression remains constant in our experimental conditions ( paragraph 2.7).

  1. Paragraph 2.9: Please provide detailed informations about the antibodies used in the experiments

Immunofluorescence  study and related Figure  are not presented  in the R1 version. Due to submission time shortage, in the absence of new experiments giving confirmed and convincing data, we decided not to present immunofluorescence experiments in R1 version.

  1. results section: The authors used 3% o2. The impact of the o2 concentration might be different in regard to the oxygen content. Do the others have data of 1% O2 experiments?

In our previous work we reported that low O2 concentrations (1.5–5%) were applied during the first phase of culture erythropoiesis ex vivo 1) enhancing of the amplification of erythroid progenitors, 2) acceleration of their proliferation and differentiation towards erythroid precursors (PMID: 19413727). These data are in line with a number of studies showing an increased number of erythroid progenitors and erythroid colony size in human bone marrow and CB as well as murine fetal liver and bone marrow progenitors in cultures at low O2 concentrations (PMID: 730784 PMID: 4054250 PMID: 1544397 PMID: 1627798 PMID: 3493045 PMID: 7138788). At 1% O2, the production and amplification of BFU-E seem to be enhanced, while their commitment toward CFU-E and further is accomplished at higher O2 concentrations (PMID: 9328456). Higher O2 concentrations (3-5%) are needed for the expansion and differentiation of committed myeloid progenitors (colony-forming cells) PMID: 15342936).These references were cited in the R1 version (discussion section, lines 474-481)

  1. Figure 2: the figure legend is partialy incomplete. Please add informations inside brackets

 Figure legend for Figure 2 is now completed in R1 version.

Reviewer 3 Report

Labat et al, using an in vitro model of erythropoiesis from CD34+ cells, clearly demonstrates that HIF1alpha, HIF2alpha and NOTCH play a crucial role in the early erythroid development. In this context authors demonstrated a functional crosstalk between HIF and NOTCH signaling. The paper is well written, experiments well designed and results clearly presented.

I have just minor points related to images:

Fig2: please use log scale in the y-axis to show cell expansion

Fig 3 and Fig4a: please cut and enlarge the lower part of the y axis to evidence differences between groups

Fig. 5 a and b: are graphs a and b referring to a particular sub-population, ie CD71+GPA+ cells? Please specify in the legend.  As most D8 cells in shHIF1 and shHIF2 are probably non-erythroid, the different expression could be related to a different cell fate, these results should be discussed also considering this.

Fig S1: please move the D0 NT CD34+ cells to the left of the figure

Fig S2: please indicate significance. Moreover it would be interesting to investigate erythroblast precursors at day 8 in NT+DAPT vs NT samples, if there are differences, would it be possible to show the scatter plots CD71 vs GPA of them?

Some typos throughout the text that can be corrected.

Author Response

Fig2: please use log scale in the y-axis to show cell expansion

 It is changed in the R1 version of the manuscript.

Fig 3 and Fig4a: please cut and enlarge the lower part of the y axis to evidence differences between groups

 The figures were modified following the reviewer’s suggestion in R1 version.

Fig. 5 a and b: are graphs a and b referring to a particular sub-population, ie CD71+GPA+ cells? Please specify in the legend.  As most D8 cells in shHIF1 and shHIF2 are probably non-erythroid, the different expression could be related to a different cell fate, these results should be discussed also considering this.

 At the  figure 5a is presented the expression of Notch2 receptor in the cells  harvested at the beginning  (Day 3, D-3), Day 4 (D 4) and Day 8 (D 8) of erythroid cultures propagated from non-transduced CD34+ cells as well as transduced CD34+ cells with shHIF1α or with shHIF2α at 20 and 3 % O2 .This is now specified in R1 version. Also, the fact that the most D8 cells in shHIF1 and shHIf2 are probably non-erythroid in the light of different NotchR expression is discussed in the results ( lines 426-429) and in the discussion section (lines 545-550).

Fig S1: please move the D0 NT CD34+ cells to the left of the figure

-It is modified in the R1 version.

Fig S2: please indicate significance. Moreover it would be interesting to investigate erythroblast precursors at day 8 in NT+DAPT vs NT samples, if there are differences, would it be possible to show the scatter plots CD71 vs GPA of them?

The significances were added.  Detailed statistical analysis showed that there is significantly more erythroblast precursors at Day 8 in the condition NT+ inhibitor DAPT versus NT condition in culture at  physiological 3% O2 as well as at 20% O2. This is now presented at Figure 4a. Corresponding scatter (dot plot) CD71 vs GPA were added as a part of Figure S3 ( Figure S3a).This finding  is accorded with the results showing that inhibition of Notch signaling pathway stimulated conversion of early erythroid progenitors into mature progenitors CFU-E (Figure 3c)  and accelerated the occurrence of GPA positive cells (Figure 4a). Altogether, these results reveal that Notch inhibition stimulates the erythroid differentiation. These conclusions are mentioned clearly in the R1 version (results section 3.3; discussion section: lines 525-529)

Some typos throughout the text that can be corrected.

- It is done in R1 version.

Round 2

Reviewer 1 Report

The authors are quite receptive to the comments and suggestions and have modified the manuscript to their best efforts. They have now included additional experimental results, have discussed the concerns raised, and have modified the manuscript in every section accordingly. With these the manuscript has improved significantly. I recommend the acceptance of the manuscript.